# Advances in the Preparation of Tough Conductive Hydrogels for Flexible Sensors

**DOI:** 10.3390/polym15194001

**Published:** 2023-10-05

**Authors:** Hongyao Ding, Jie Liu, Xiaodong Shen, Hui Li

**Affiliations:** 1College of Materials Science and Engineering, Nanjing Tech University, Nanjing 210009, China; hongyaoding@njtech.edu.cn (H.D.);; 2Key Laboratory for Light-Weight Materials, Nanjing Tech University, Nanjing 210009, China

**Keywords:** hydrogels, toughness and conductivity, flexible sensors

## Abstract

The rapid development of tough conductive hydrogels has led to considerable progress in the fields of tissue engineering, soft robots, flexible electronics, etc. Compared to other kinds of traditional sensing materials, tough conductive hydrogels have advantages in flexibility, stretchability and biocompatibility due to their biological structures. Numerous hydrogel flexible sensors have been developed based on specific demands for practical applications. This review focuses on tough conductive hydrogels for flexible sensors. Representative tactics to construct tough hydrogels and strategies to fulfill conductivity, which are of significance to fabricating tough conductive hydrogels, are briefly reviewed. Then, diverse tough conductive hydrogels are presented and discussed. Additionally, recent advancements in flexible sensors assembled with different tough conductive hydrogels as well as various designed structures and their sensing performances are demonstrated in detail. Applications, including the wearable skins, bionic muscles and robotic systems of these hydrogel-based flexible sensors with resistive and capacitive modes are discussed. Some perspectives on tough conductive hydrogels for flexible sensors are also stated at the end. This review will provide a comprehensive understanding of tough conductive hydrogels and will offer clues to researchers who have interests in pursuing flexible sensors.

## 1. Introduction

Hydrogels, composed of chemical or physical crosslinked polymer networks and large amounts of water, have gained significant attention in recent years. Due to their unique features such as environmental responsiveness, flexibility and biocompatibility, hydrogels have been widely used in the applications of tissue engineering, drug delivery, soft robotics, energy storage, intelligent actuators, wearable electronics, etc. [1,2,3,4,5,6]. For example, some hydrogels could revolutionize tissue engineering by providing a versatile platform for creating functional and biomimetic scaffolds that have the ability to mimic the mechanical properties of natural tissues and can further allow for better integration with surrounding tissues and improve cell viability. Various kinds of hydrogels have been developed with tailored properties to fit unique tissue engineering for the cartilage, bone and blood vessels [7,8]. Soft robots often require materials with the ability to deform and recover their shapes, and hydrogels offer excellent compliance and flexibility. Various hydrogel-based actuators and structural components have been developed and used in soft robotic systems, which can perform complex movements and interactions under specific external stimuli [9,10]. Hydrogels are also the proper candidates for constructing electronic devices with the features of bend, stretch and conformity to irregular surfaces, which can be worked into flexible electronics such as stretchable electrodes, sensors and interconnections [11,12].

As an essential application branch, flexible sensors based on hydrogels with various sensing mechanisms have attracted much attention in recent years and are widely used for real-time health monitoring, flexible energy storage devices and stretchable electronic circuits [13,14,15]. However, many traditional natural and synthetic hydrogels show the mechanical characters of weakness and brittleness. For example, fracture stress (σ_b_), fracture strain (ε_b_) and fracture energy are generally less than 100 kPa, 100% and 100 J/m^2^, respectively, which is relative to the inherent inhomogeneous network structures or the lack of effective energy dissipation mechanisms [16,17,18]. Many efforts have been made in recent years with the aspects of hydrogel substrate, crosslinking ways and network structure to increase energy dissipation [19,20,21,22,23]. This improvement in hydrogels’ mechanical performance makes it possible for their employment in the field of flexible sensing.

Conductive character is also critical for hydrogel sensors, and it can directly influence sensing performance. Hence, it is crucial to find suitable methods to integrate proper conductivity matters into gel networks and thus impart the appropriate electronic properties [9,24,25,26]. Therefore, a gel’s performances can greatly influence the usage of constructed devices, especially the trade-off of mechanical and electrical properties [27,28]. In this review, some recent advances in tough conductive hydrogels for flexible sensors are summarized and discussed. First, the design principles of tough hydrogels and the strategies for fabricating conductive hydrogels are discussed. Then, diverse tough conductive hydrogels are presented and discussed. Additionally, recent advancements in flexible sensors assembled with different tough conductive hydrogels as well as variously designed structures and their sensing performances are demonstrated in detail. Finally, the current challenges and future perspectives of tough conductive hydrogels for flexible sensors are discussed based on the latest achievements. It is expected that this review will help researchers gain an understanding of tough conductive hydrogels for flexible sensing and also provide some advice to those whose want to pursue a career in this emerging area.

## 2. Requirements for Fabricating Tough Hydrogels

Tough hydrogels with satisfactory mechanical properties are suitable for the fabrication of load-bearing materials. In general, two kinds of effective approach are commonly adopted to devise tough hydrogels. The first one is to construct a homogeneous hydrogel network structure, which can effectively avoid stress concentration on network defects, such as in slide-ring hydrogels or tetra-PEG hydrogels [29,30,31]. Another method for fabricating tough hydrogels is to design an effective energy dissipation mechanism inside a gel matrix at different scales from micro to macro, including the destruction of polymer chains, the introduction of reversible crosslinking points and the conformational transformation of polymer chains or crosslinking points, as well as the opening and breaking of macroscopic reinforcing phases [32,33,34,35,36,37,38]. Under the premise of these methods, a variety of tough hydrogels have quickly emerged, such as double-network (DN) hydrogels and tough hydrogels reinforced with hydrogen bonds or electrostatic interactions. During the loading process, the special network structure can dissipate a large amount of energy due to the destruction of chemical bonds or associative interactions [39,40,41,42,43], which is beneficial for keeping hydrogels intact. The second scheme is mainstream for constructing tough hydrogels, for which the synergistic effect of several energy dissipation mechanisms is also often used.

The selected tough hydrogel strategy should be regulated according to mechanical parameters such as σ_b_, ε_b_, Young’s modulus and recoverability in specific applications [44]. For example, the σ_b_ of a hydrogel determines the maximum load-bearing capacity before failure, which is crucial in applications where the hydrogel needs to withstand extra force, such as tissue engineering scaffolds or medical devices [8]. The elongation of the constructed hydrogel will determine its deformation capacity, and high elongations are desirable in applications such as soft robotics or flexible electronics, which need repeated stretching or deformation [45]. The modulus is related to a gel’s ability to maintain its shape and provide structural support. Hydrogels used in wearable electronics require a low modulus similar to that of the skin [46], and hydrogels employed in tissue engineering or drug delivery systems need a high modulus to keep mechanical stability and shape retention [47]. Hydrogels with excellent recoverability are desirable in applications involving cyclic loading or repeated deformation such as wearable electronics and biomedical adhesives [48]. Overall, properly regulating these mechanical parameters allows hydrogels to satisfy specific demands on different applications, enabling their successful implementation in various fields.

## 3. Design Elements for Conductive Hydrogels

Hydrogel flexible sensors typically work based on their changes in physical or chemical properties in response to specific stimuli, which can be detected by auxiliary instruments, allowing these sensors to provide information about the target analytes or environmental conditions [49,50,51,52,53]. Conductive hydrogels are the essential ingredients for constructing flexible sensors, and the conductive property can directly affect the sensors’ assembly methods, mechanisms and sensing performances [13,54,55,56]. Therefore, the first design element for fabricating a sensor is to build the suitable conductive hydrogel for the targeted assignment.

Generally, conductive hydrogels are a promising class of materials, offering the combination of flexibility and electrical conductivity. The construction of conductive hydrogels mainly includes the selection of gel network compositions (provide support) and conductive components (provide electrical conductivity) [57]. The rapid development of conductive hydrogels has gained significant attentions in recent years, and there have notable achievements in this area. Commonly, conductive hydrogels can be prepared by incorporating conductive polymers, metal/carbon-based materials or ionic salts into the elastic 3D hydrogel matrix [58,59,60,61,62,63], which can be roughly divided into two categories, including electronic conductive hydrogels (Figure 1A) and ionic conductive hydrogels (Figure 1B) [64]. Because of the difference in conductive materials, the conductive hydrogels have different network structures and conductive mechanisms.

### 3.1. Electronic Conductive Hydrogels (ECHs)

The first kind is the electronic conductive hydrogels (ECHs), which is prepared by combining or hybridizing hydrogel networks with conductive materials (graphene, carbon nanotubes, metal nanoparticles or nanowires, conductive polymers) [65,66,67,68,69]. The conductive fillers can form conductive paths or networks inside the gel matrix, which endow the hydrogels with conductivity [70,71]. Some conductive materials can be used directly to construct hydrogels with high conductivity. For example, Xu et al. fabricated physically crosslinked graphene hydrogels with graphene nanosheets reduced via a one-step hydrothermal method, showing a high electrical conductivity of 0.5 S·m^−1^ [72]. However, the formed conductive network can easily be broken under low strain, resulting in the fracture of the gel matrix and the loss of conductivity. Therefore, these ECHs commonly show poor flexibility and stretchability, restricting their practical application severely. In order to address this problem, researchers prepared ECHs by incorporating conductive materials into a stretchable polymer matrix, obtaining the conductive hydrogels with high flexibility and stretchability. Turng et al. reported a nanocomposite hydrogel prepared with polyacrylic acid (PAA) and reduced graphene oxide (rGO) via mussel-inspired chemistry [73]. The introduction of the rGO not only imparts the gel with high stretchability over 600% and σ_b_ of 400 kPa, but offers good conductivity in sensing applications (Figure 2A). Xu et al. fabricated the ECHs based on the hybrid assembly of polymeric nanofiber networks using polypyrrole, aramid nanofibers and polyvinyl alcohol [74]. The obtained composite hydrogels have excellent conductivity, up to 80 S/cm, and a remarkable σ_b_ of 9.4 MPa (Figure 2B). Chong et al. prepared the polyacrylic acid/poly(3,4−ethylenedioxythiophene):polystyrene sulfonate (PAA/PEDOT:PSS) ECHs via a template-directed assembly method that enables the formation of a high conductive nanofibrous network inside the highly stretchable gel matrix [75]. The obtained ECHs possess a high conductivity up to 247 S/cm and good mechanical properties with σ_b_ of 100 kPa and ε_b_ of 610% (Figure 2C). Overall, these conductive materials usually have high conductivities due to the special electronic conductive mechanisms, but they also have some problems such as an uneven dispersion of conductive fillers, non-biocompatibility, poor transparency and low strain limitations.

### 3.2. Ionic Conductive Hydrogels (ICHs)

Another candidate is ionic conductive hydrogels (ICHs), which consists of the polymer networks and free ions. The ionic conductivity of these hydrogels is achieved through the incorporation of various ion-containing materials into a polymer network [76,77,78]. The water environment allows for the movement of ions within the hydrogel network, enabling them to act as conductive pathways [79,80]. The ion-containing substances used for constructing conductive hydrogels mainly include inorganic salts, ionic liquids and polyelectrolytes containing counterions.

The inorganic salts or ionic liquids are commonly used to fabricate conductive hydrogels because of the wide variety and easy availability, especially halogenated salts of metal ions [81,82,83]. There are two ways to introduce inorganic salts into hydrogel systems. The first is to add inorganic salts into the hydrogel precursor solutions, which then polymerizes or crosslinks to form solid-like gel materials. For example, Ding et al. fabricated a tough and conductive ionic composite hydrogel with a semi-interpenetrating network structure by introducing carboxymethyl chitosan and sodium chloride into a crosslinked polyacrylamide network [84]. The gels possess good mechanical properties due to the high content of hydrogen bonding formed in the gels’ network, with a σ_b_ of 430 kPa and a ε_b_ of 1100%; simultaneously, high conductivity up to 6.44 S/m is obtained in this system (Figure 3A). Another method is the post-introduction method, which is to immerse the obtained as-prepared gels into inorganic salt solutions or ionic liquids. For example, Chen et al. first prepared the supramolecular hydrogels by incorporating bentonite (BT) via the strong cellulose–BT coordination interactions [85]. Then, the obtained neutral cellulose–BT hydrogels were immersed into LiCl aqueous solutions to obtain the ionic conductive gels. The final products have excellent mechanical properties with a compressive strength of 3.2 MPa, and have excellent conductivity up to 89.9 mS/cm (Figure 3B). Kong et al. prepared the strong wood hydrogels with cellulose nanofibers and polyacrylamide, which turned into ionic conductors after being dropped into potassium chloride solutions, showing the high conductivity of 0.05 S/m [77]. Zheng et al. designed the conductive gels by dropping the zwitterionic hydrogels prepared with a structurally ameliorated sulfobetaine monomer into the 1-ethyl-3-methylimidazolium tetrafluoroborate, and the resulted gels possess good conductivity, up to 3.48 S/m [86]. This method is simple and effective. However, the ions inside the gel networks are easily affected by changes in the external environment. For example, when the ionic conductive gel is placed in an aqueous condition, the ions in the network will migrate out of the gel’s surface, thus affecting the ion concentration and changing the conductivity of the gel conductor. Therefore, it is necessary to pay more attention to the environment before actual usage.

Other ion-containing materials include the natural and synthetic polymer polyelectrolytes. The mobile ions inside the gel network are derived from the counterions belonging to groups on the polymer chains. For example, Cui et al. designed the biomimetic hydrogels with a strain-stiffening property by embedding highly swollen poly(acrylate sodium) microgels into the polyacrylamide matrix, which exhibit unique mechanical properties with a σ_b_ of 0.99 MPa and a low modulus of 113 kPa [87]. The obtained poly(acrylate sodium) microgels have a large amount of sodium ions, endowing the gels with good conductivity up to 2.2 mS/cm (Figure 3C). The mobile ions are trapped, surrounding the group sites on the polyelectrolyte polymer chains, which avoid the adverse diffusion of ions in aqueous conditions. The properties of ionic conductive hydrogels can be tailored by adjusting the composition and structure of the polymer network, as well as the type and concentration of the ion-containing substances [88,89,90]. This allows for the optimization of their appearance, conductivity, mechanical property and biocompatibility for specific applications. It should be noted that the free ions inside the gel matrix can be influenced by the external aqueous environment because of the variation of the ion concentration derived from the polyelectrolyte effect [91].

Besides the advances stated above, compound conductive substances have also been investigated for fabricating conductive hydrogels. Generally, the resultant hydrogels have better conductivity than the single ECHs and ICHs. For example, Zhang et al. prepared the polyacrylamide/sodium alginate (PAM/SA) double-network hydrogels, in which the polypyrrole nanospheres, sodium ions and ferric ions were distributed. The obtained hydrogels not only have good mechanical properties due to the synergies of multiple energy dissipation mechanisms with σ_b_ of 600 kPa and ε_b_ of 800%, but also possess a high conductivity of 10 S/m (Figure 4A) [92]. Liang et al. designed the tough conductive composite hydrogels via a two-step method, and the resultant gels contain a polyacrylamide polymer network, dispersed acidified single-walled carbon nanotube and tin(IV) chloride/tin(II) chloride ions [93]. The combination of the electronic and ionic conductive materials makes the gels possess special network structures, resulting in a remarkable conductivity up to 13.47 S/m (Figure 4B). The obtained hydrogels also perform good mechanical properties with an ε_b_ of 1300%. The synergistic effect of various components has different impacts on the performances of these compound conductive hydrogels. We have concluded the conductivity of some typical hydrogels as shown in Table 1. Generally, the conductivity of the ECHs is also better than that of ICHs due to the special electronic conductive mechanism. The compound conductive substances selected in fabricating conductive hydrogels will be a good choice for researchers to pursue high conductivity in this region. Also, the performance balance of the conductive hydrogels should be paid more attention, especially the mechanical and electrical property, which is crucial for the practical applications as flexible sensors.

## 4. Applications of Tough Conductive Hydrogels for Flexible Sensors

The improvements of the mechanical and electronic properties broaden the applications of conductive hydrogels. The applications of tough conductive hydrogels are diverse and span across various fields, especially in flexible electronics [44,104,105]. Commonly, conductive hydrogels are used to fabricate flexible hydrogel sensors due to their excellent flexibility and stretchability. The sensing performances of the constructed sensors are not only related to the hydrogels’ structures and properties, but also closely related to the sensors’ configurations and types [71,106]. Conventionally, the conductive hydrogels can be made into resistive-based hydrogel sensors and capacitive-based hydrogel sensors in terms of the assembly methods. In this section, the multifunctional conductive hydrogels for resistive and capacitive sensors will be emphasized and their recent advances will also be summarized.

### 4.1. Resistive-Type Hydrogel Sensors

Resistive-based hydrogel sensors are the common type of flexible electronics that utilize changes in electrical resistance to measure strain or deformation induced by external forces. The conductive hydrogel is directly connected to the circuit as a conductor, and a testing machine, such as a multimeter or an electrochemical workstation, will be connected in parallel to both ends of the hydrogel to test the real-time resistances under different strains [107,108].

The constructed sensors can be used in many potential applications. For example, Tang et al. fabricated the tough and conductive chitosan/poly(vinyl alcohol) hydrogels/phytic acid (PA)/boric acid (BA) hydrogels (CS/PVA−PA−BA) [94]. They have satisfactory mechanical properties, with an ε_b_ of 1070% of and a σ_b_ of 830 kPa. The free hydrogen ions (H^+^) produced from PA and BA endow the hydrogels with good ionic conductivity up to 5.3 S/m, resulting in the fabrication of strain sensors with the conductors. The constructed hydrogel sensors demonstrate a high sensitivity with a gauge factor of 4.6; a broadened linear strain, ranging up to 1000%; a fast response time of about 90 ms and good stability during repeatable tests (Figure 5A). Cui et al. prepared the tough and conductive hydrogels (gelatin−PAAc−MXene−Zr^4+^) formed by the synergy of MXene-activated fast initiation and zirconium ion (Zr^4+^)-induced rapid crosslinking [95]. The obtained composite hydrogels possess an anti-swelling property and have a high conductivity of 1.76 S/m that can even stabilize under water within 30 days. Therefore, it is suitable for the conductive gels to be used in sensing applications underwater. When constructed as strain sensors, they are very sensitive in detecting the bending of a human finger underwater with the gauge factors of 1.24–1.92. Simultaneously, the hydrogel sensors have good stability in the cyclic experiments. On this basis, a hydrogel-based underwater communicator was demonstrated by a self-powered wireless transmission device, in which the gel can convey different information via Morse codes. This equipment can deliver the resistance changes during deformations underwater to a decoder to translate the electrical signals into visible letters, which is of great importance once an emergency takes place for the aquanaut underwater (Figure 5B). Gu et al. designed the conductive hydrogels with PEDOT:PSS nanofibers and PVA [96]. The facile preparation method combines 3D printing and successive freeze-thawing operations. The obtained hydrogels possess a high stretchability of 300% and negligible hysteresis below 1.5%. They systematically studied multiple applications with strain sensors prepared with this conductive material. The fabricated wearable sensors can detect the carotid artery pulse waveforms and heart rate, and can monitor the subtle human joint motions (blinking, swallowing and frowning). In order to record and interpret the complex motions accurately, they also developed a five-pixel wearable sensor that can precisely detect the hand gestures of American Sign Language after signal plotting by the record of the resistance changing from all the five channels. The authors also integrated the hydrogel strain sensors into a two-finger pneumatic soft gripper to explore the sensing functionality. The resultant soft gripper can autonomously record the bending angles of an arm controlled by the pneumatic pressure, and the gripper can even obtain the sizes of grasping objects through the strain changing by varying the pneumatic pressure. A sensory robotic system with four channels was further developed with this conductive hydrogel, which can cause the robot to move through a labyrinth-like trajectory (Figure 5C). The specific information such as the sensing parameters and application area of some selected hydrogel-based resistive sensors are summarized in Table 1.

### 4.2. Capacitive-Type Hydrogel Sensors

Another way to prepare flexible sensors is to compose a capacitive mode with a sandwich structure assembled with two layers of conductive hydrogels and a layer of dielectric material, which can utilize the properties of the constructed hydrogel blocks to measure changes in capacitance [1]. Evidently, the capacitance of the constructed structure is the ability to store the electrical charge, and it can be easily influenced by various ingredients such as the external pressure, humidity, chemicals or charged matters [109,110,111]. The hydrogel-based capacitive sensors have many potential applications in a wide range of fields. For example, Li et al. introduced a strategy to prepare the tough and conductive PAM−CaCl_2_ polymer hydrogels according to the spatial confinement induced by hydration [97]. The special network structures provide the gels with unique mechanical properties, with a σ_b_ of 1.5 MPa, an ε_b_ of 2400% and a low modulus of 20 kPa. The hydrogels also display a negligible hysteresis of 0.13% during loading–unloading cycles with a fixed strain of 1000%, indicating the excellent self-recovery ability. The free ions of Ca^2+^ and Cl^−^ inside the network endow the gel with high electronic conductivity. The combination property indicates that the gel can be very suitable for the fabrication of strain sensors. A sandwich structure capacitive strain sensor was thereby fabricated with the conductive and tough hydrogel. As expected, the strain sensors show high sensitivity in detecting the finger bending behaviors and possesses excellent stability during cyclic tests. The constructed capacitive hydrogel sensors have the ability in recognizing Morse code, which can be used to design the signals of dots and dashes to delivery various massages (Figure 6A). The geometry of the conductive gel in a constructed device has a great influence on the sensor’s performance. Chen et al. developed the PVA/PANI hydrogels through the physical crosslinking strategy [98]. The resultant hydrogels possess the flexibility and conductivity. One surface of the hydrogel conductor has some regulated reliefs fabricated via the template method. The obtained hydrogels with special surface morphology were adopted to fabricate the capacitive pressure sensors. The size of the designed reliefs of hydrogels has great influence on the sensitivity, and the introduction of the reliefs improves the sensor’s sensitivity to 7.7 kPa^−1^ that are much higher than other sensors prepared without reliefs. In addition, the lower thickness of a hydrogel layer will also result in a higher sensitivity value in the capacitive pressure sensor. The constructed hydrogel sensors can monitor various small pressures in daily life such as the blowing or the process of writing with a brush (Figure 6B). Suo et al. fabricated an ionic hydrogel conductor using the PAM and NaCl [55]. The obtained conductive gel has high transparency and stretchability. They equipped the capacitive sensors using the conductive gels and VHB tapes, and they can detect human motions such as finger bending with stable and notable signals. A sheet of distributed sensors was designed to demonstrate the capability of detecting the appointed locations (Figure 6C). Therefore, with the development of the sensing field, the applications of the hydrogel sensors have been broadened rapidly in recent years. The sensing functionality has become more and more diverse with the improvement of the combination properties of the conductive hydrogels and the complex structural design of the desired devices. The specific parameters of some selected hydrogel-based capacitive sensors are also listed in Table 1. Compared with the resistive-based hydrogel sensors, the capacitive-based ones have a higher sensitivity due to the special structures. Compared to other sensing technologies in signal detections, these hydrogel sensors demonstrated above have many unique merits, such as high sensitivity, wide detectable signal range, flexibility, biocompatibility, fast response time, low power consumption or low cost-effective fabrication, making them attractive in a wide range of applications.

## 5. Conclusions and Outlook

Tough and conductive hydrogels have a bright further in the design of flexible sensors due to their flexible nature and special network structures. This review firstly demonstrates the representative tactics to construct tough hydrogels and strategies to fulfill conductivity, and then presents diverse tough conductive hydrogels which were recently reported. Additionally, recent advancements in flexible sensors assembled with different tough conductive hydrogels as well as various designed structures and their sensing performances are demonstrated in detail. Applications of these hydrogel-based flexible sensors with resistive and capacitive modes are discussed simultaneously.

In the development of hydrogel flexible sensors, many factors, including the choice of conductive hydrogel material, the geometric structure design, the assembly method and the integrated development of functions, should all be comprehensively considered [112,113]. First of all, it is a challenge to achieve the trade-off between electrical conductivity and mechanical property for the desired tough conductive hydrogels. Although the hydrogels are commonly soft and flexible, the addition of other conductive matters may weaken their mechanical integrity. The ideal conductive hydrogel for flexible sensors usually possess high conductivity, elongation and strength, as well as low modulus and hysteresis. Therefore, researchers should make more of an effort to optimize the compositions and structures of the desired conductive hydrogels to achieve the balance.

The long-term stability of the tough conductive hydrogels should also be concerned during the processes of design and usage. Many hydrogel sensors are used under energized conditions, and electrochemical reactions occur easily during use [114,115]. The compositions of the gels in the circuit will be changed under long-term experiments, which will directly affect their comprehensive properties. It should be noted that the developed flexible sensors could be used in a variety of harsh environments. More attention should be placed on the aspects of gels’ performances in terms of some key points such as low temperature resistance, water retention and swelling resistance. For example, some inorganic salts were selected to be added into the gel systems to improve the low temperature resistance and water retention [115,116,117,118]. The bionic structures were selected to prepare the gel networks to improve their water retention. Although there has been some progress, researchers are expected to make more of an effort to develop new technologies and concepts on improving the long-term stability and durability of the sensors.

The improvement of the sensitivity is of importance for the performance and accuracy of the constructed flexible sensors [105]. It should be noted that the specific ways to increase sensitivity should be appropriately regulated depending on the sensor type and practical applications. By considering these factors and integrating them into targeted requirements, the sensitivity of flexible sensors could be evidently improved, which can lead to more accurate and reliable measurements. For example, choosing the conductive matter is essential for the sensitivity [119], and the electronic conductive materials can provide better electrical performances than the ionic ones. Structural designs incorporating micro- or nanostructured patterns on the sensors’ surfaces can increase the effective surface area and thus improve the sensitivity [120,121]. Designing architecture geometries such as special three-dimensional hollow structures and aspect ratios can improve sensitivity to small changes in pressure or strain, and the methods are commonly combined with advanced fabrication techniques like 3D printing [122,123,124,125,126,127]. The integration of multiple sensors, including integrated multiple sensing modalities or employed sensor arrays, can also boost the sensitivity by providing redundancy and a cross-validation of measurements [128,129,130].

Overall, tough conductive hydrogels offer great potential for developing flexible sensors used in the fields of healthcare, robotics, environmental monitoring or energy systems. Ongoing research and advancements in hydrogel materials, fabrication techniques and integration methods will continue to expand the capabilities and applications of hydrogel-based flexible sensors, which would lead to great progress in technology, human wellbeing and sustainability.

## Figures and Tables

**Figure 1 polymers-15-04001-f001:**
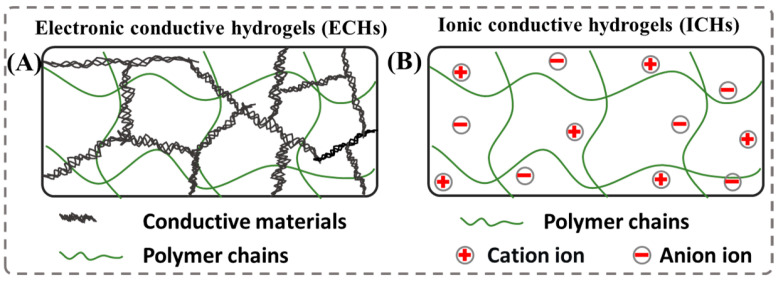
Schematic drawing of (**A**) electronic conductive hydrogels (ECHs) and (**B**) ionic conductive hydrogels (ICHs).

**Figure 2 polymers-15-04001-f002:**
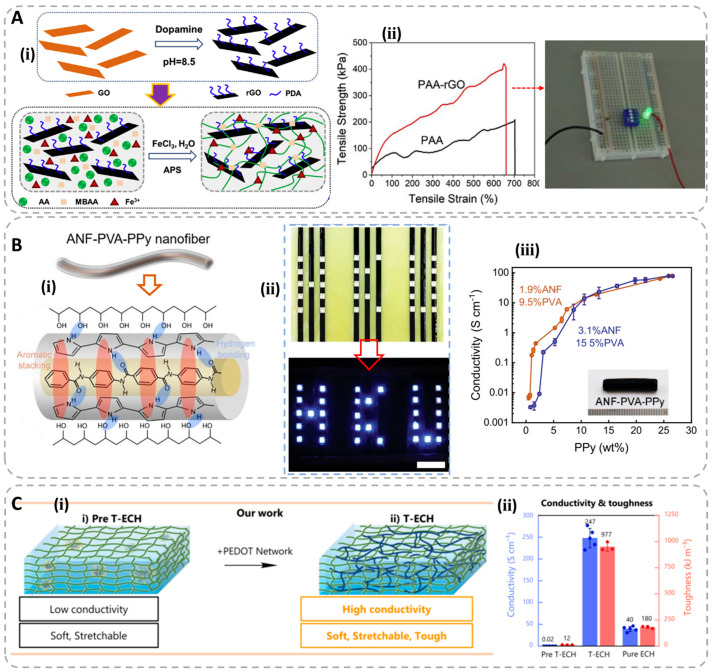
(**A**) Illustration of the preparation of PAA/rGO hydrogels (**ⅰ**); tensile mechanical curves and illustration of the gel’s conductivity (**ⅱ**). Reproduced with permission from [73]. (**B**) Illustration of ANF/PVA/PPy hydrogels (**ⅰ**); display of the conductivity of hydrogels (**ⅱ**); conductivity values of the hydrogels with different ppy content (**ⅲ**). Reprinted from ref. [74]. (**C**) Illustration of the fabrication of PAA/PEDOT:PSS hydrogels (**ⅰ**); conductivity and toughness of different hydrogels (**ⅱ**). Reprinted from ref. [75].

**Figure 3 polymers-15-04001-f003:**
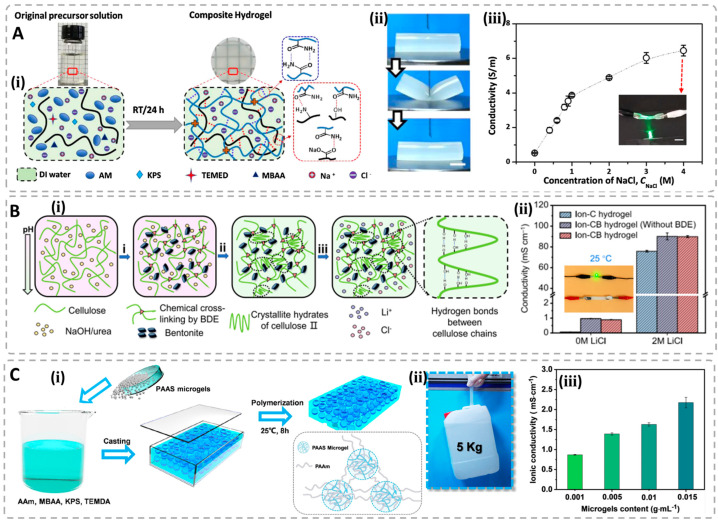
(**A**) Illustration of the preparation of PAM/CMC/NaCl composite hydrogels (**ⅰ**); diagrams of hydrogel before and after loading (**ⅱ**); conductivity of hydrogels with various compositions (**ⅲ**). Reproduced with permission [84]. (**B**) Preparation processes of the ionic conductive cellulose–BT hydrogels (**ⅰ**) and conductivity of different hydrogels (**ⅱ**). Reprinted from ref. [85]. (**C**) Illustration of the preparation of PAAS microgel/PAAm hydrogels (**ⅰ**); picture showing a gel withstand the 5 kg weight (**ⅱ**); conductivity of various hydrogels with different compositions (**ⅲ**). Reproduced with permission from [87].

**Figure 4 polymers-15-04001-f004:**
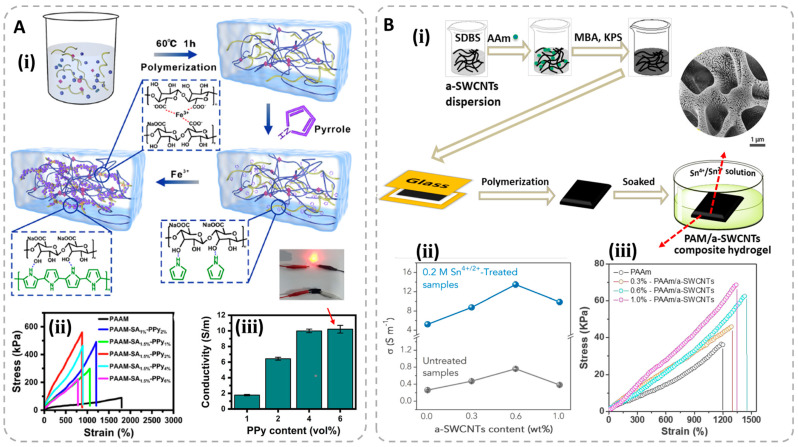
(**A**) Illustration of the preparation of PAM/SA/polypyrrole/ferric ions composite hydrogels (**ⅰ**); tensile curves of various hydrogels (**ⅱ**); conductivity of different hydrogels with various ppy content (**ⅲ**). Reproduced with permission from [92]. (**B**) Preparation of PAAm/a−SWCNT/Sn^4+^/Sn^2+^ conductive composite hydrogels (**ⅰ**); conductivity of various hydrogels with different compositions (**ⅱ**); tensile curves of various conductive hydrogels (**ⅲ**). Reproduced with permission from [93].

**Figure 5 polymers-15-04001-f005:**
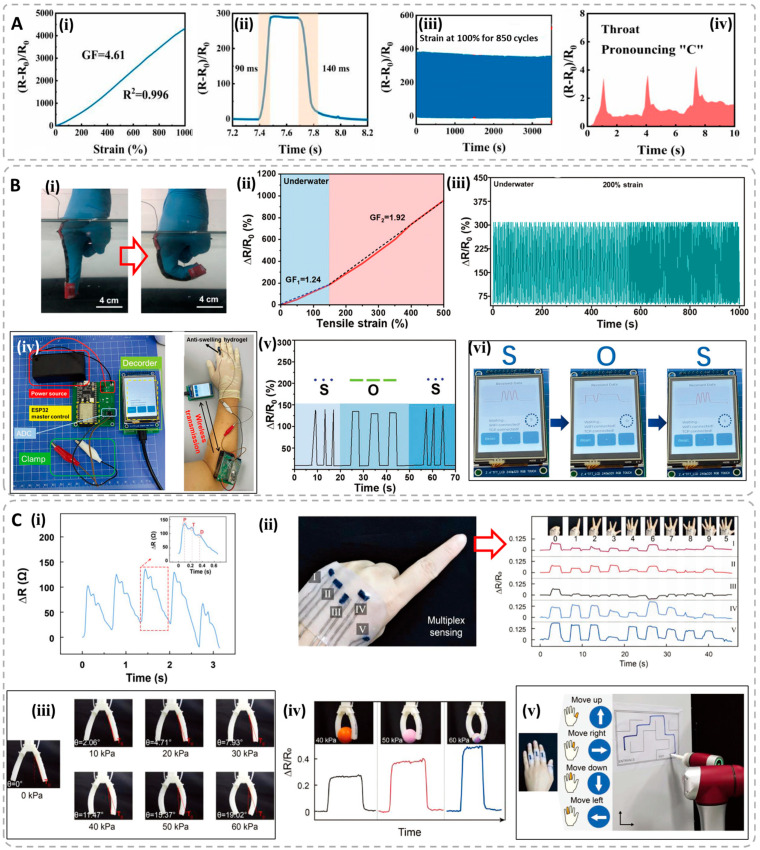
(**A**) Relative resistance changes and the gauge factor of the hydrogel sensor under different tensile strains (**ⅰ**); response and recovery time of the sensor under a strain of 50% (**ⅱ**); relative resistance changes of the hydrogel sensor in the loading–unloading cycles with the fixed strain of 100% (**ⅲ**); real-time monitoring of human activities (**ⅳ**). Reproduced with permission from [94]. (**B**) The applications of the gelatin−PAAc−MXene−Zr^4+^ hydrogel sensors. Illustration of the working model of the gel sensor underwater (**ⅰ**); the relative resistance changes of the gel at different strains underwater (**ⅱ**); the resistance changes of the gel during loading–unloading cycles at a fixed strain of 200% underwater (**ⅲ**); a photograph showing the components of the self-powered wireless transmission device (**ⅳ**); the signals of different Morse codes by controlling the deformation time (**ⅴ**); message displayed on the screen of the decoder by translating the signals into visible English letters (**ⅵ**). Reproduced with permission from [95]. (**C**) The applications of the PEDOT:PSS−PVA hydrogel sensors. Monitoring of human carotid artery pulse (**ⅰ**); a wearable electronic skin mounted on the back of the hand for gesture recognition and the relative changes in resistance of five channels with different gestures (**ⅱ**); two-finger pneumatic soft gripper derived by different pressure (**ⅲ**); soft gripper can detect the grasping process using spherical objects with different radiuses (**ⅳ**); a constructed robotic system can remotely control an industrial robot to move through a targeted labyrinth (**ⅴ**). Reproduced with permission from [96].

**Figure 6 polymers-15-04001-f006:**
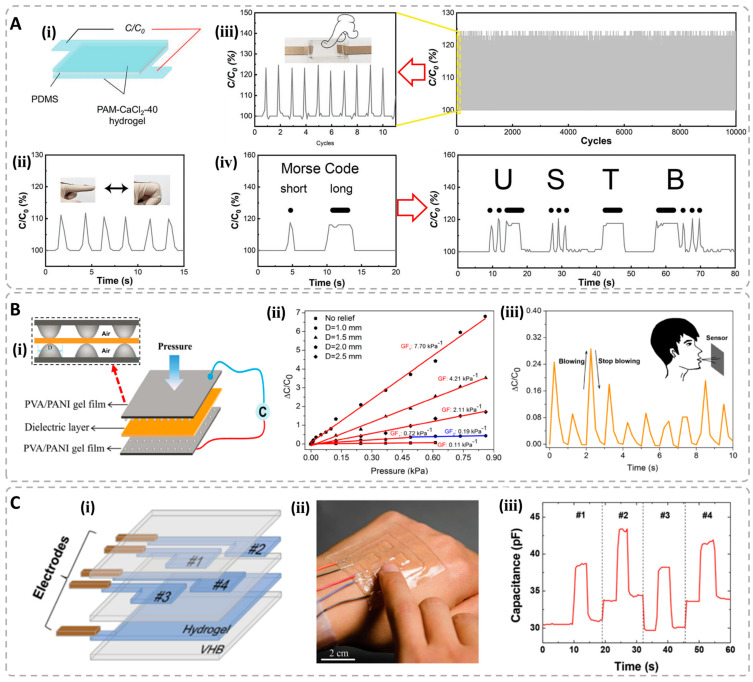
(**A**) Illustration of the constructed capacitive strain sensor (**ⅰ**); capacitance variation for detecting finger bending (**ⅱ**); press cycling experiments of the capacitive strain sensor (**ⅲ**); the construction of the machine worked with Morse code using the capacitive signals by controlling the pressing time to achieve dots and dashes, and the certain Morse code signal detected by the capacitive strain senso (**ⅳ**). Reproduced with permission from [97]. (**B**) Illustration of the cross-section area of the pressure sensors (**ⅰ**); capacitance changes of the pressure sensor without reliefs and the pressure sensors containing reliefs with different size (**ⅱ**); monitoring of pressure induced by slight blowing (**ⅲ**). Reproduced with permission from [98]. (**C**) Illustration of an array constructed with the PAM/NaCl hydrogels (**ⅰ**); a picture of the sensor array (**ⅱ**) and its practical application used for the location detecting (**ⅲ**). Reproduced with permission from [55].

**Table 1 polymers-15-04001-t001:** The comparison of conductive hydrogels as flexible sensors.

Hydrogel Materials	Conduction Type	Conductivity	Sensing Type	Gauge Factor	Sensing Range	Fitting Relation	Application Field	Ref.
ANF−PVA	Electronic	80 S/cm	Resistive	N/A	N/A	N/A	Bioelectronics	[74]
PAA/PEDOT:PSS	Electronic	247 S/cm	Resistive	N/A	N/A	N/A	Bioelectronics	[75]
CMC/PAM/NaCl	Ionic	6.44 S/m	Resistive	0.104–0.214	0.5–800%	Piecewise linearity	Wearable skin	[84]
Cellulose/bentonite	Ionic	89.9 mS/cm	Resistive	N/A	N/A	N/A	Wearable skin	[85]
MRH	Ionic	2.2 mS/cm	Resistive	2.98–6.78	0–300%	Piecewise linearity	Wearable skin	[87]
PAAM−SA−PPy NSs	Ionic/Electronic	10 S/m	Resistive	1.89–4.53	0–800%	Piecewise linearity	Wearable skin	[92]
PAAm/a−SWCNT/Sn	Ionic/Electronic	13.47 S/m	Resistive	0.3–0.8	0–100%	N/A	Wearable skin	[93]
CS/PVA−PA−BA	Ionic	5.3 S/m	Resistive	4.61	0–1000%	Linearity	Wearable skin	[94]
GPMZr	Ionic/Electronic	1.76 S/m	Resistive	1.24–1.92	0–500%	Piecewise linearity	Wearable skin/Underwater communicator	[95]
PEDOT:PSS−PVA	Electronic	N/A	N/A	4.07	0–300%	Linearity	Robotic skin	[96]
PAM−CaCl_2_	Ionic	N/A	N/A	N/A	N/A	N/A	Wearable skin/Bionic muscle	[97]
PVA/PANI	Electronic	N/A	Capacitive	1.26–7.7 kPa^−1^	0–7.4 kPa	Piecewise linearity	Wearable skin	[98]
PAM/NaCl	Ionic	N/A	Capacitive	N/A	N/A	N/A	Wearable skin/Location detecting	[55]
PDA@CNT/PAAm	Electronic	2 mS/m	Resistive	1.99–3.93	0–400%	Piecewise linearity	Wearable skin/Smart ring device	[99]
HAPAA/PANI	Ionic/Electronic	3.35 S/m	Resistive	2.6–17.9	0–1500%	Piecewise linearity	Wearable skin/Touch screen device	[50]
PAAm/LiCl	Ionic	1 S/m	N/A	N/A	N/A	N/A	Touch panel	[100]
TiO_2_/PDMAA	Electronic	N/A	N/A	N/A	N/A	N/A	Touch panel	[60]
P(AAc−co−SBMA)	Ionic	N/A	Capacitive	0.9 kPa^−1^	0–5 kPa	Linearity	Wearable skin	[101]
PMVIC−r−SAMS	Ionic	0.55 S/m	Capacitive	1.12–1.14	0–200%	Piecewise linearity	Wearable skin	[102]
PSCL	Ionic	17.1 mS/cm	Capacitive	0.14 °C^−1^	12–95 °C	Nonlinearity	Temperature sensor	[103]

N/A indicates that this information is not applicable in the reference.

## Data Availability

Not applicable.

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
