# Peer review of "Advances in the Preparation of Tough Conductive Hydrogels for Flexible Sensors"

_polymers, 2023, doi:10.3390/polym15194001_

Round 1

Reviewer 1 Report

This manuscript reports the Advances in the Preparation of Tough Conductive Hydrogel for Flexible Sensors. In my opinion, the manuscript is suitable for publication in the polymers, after the authors have addressed the following major comments.

  1. How has the rapid development of tough conductive hydrogels impacted the fields of tissue engineering, soft robots, and flexible electronics, and what specific progress has been achieved in each of these domains?

  2. Could you explain the role of the "toughening strategy" within gel networks in regulating mechanical parameters like breaking stress, elongation, modulus, and recoverability? Why are these parameters important for specific applications?

  3. What recent advancements in the preparation of tough conductive hydrogels for flexible sensors are highlighted in the review, and how do these advancements contribute to the field of soft materials?

  4. How does the review emphasize the "toughening strategy" and "conductive method" in the context of hydrogel development, and what makes these aspects significant?

  5. Can you provide examples of applications where these tough conductive hydrogels have been used as flexible sensors with resistive and capacitive modes for detecting additional signals? What advantages do these modes offer in signal detection?

  6. What are some of the current challenges associated with using tough hydrogels as flexible sensors, and how might overcoming these challenges lead to advancements in the development of soft materials for future devices?

  7. Could you elaborate on the future perspectives mentioned in the abstract? How do the authors envision the role of tough hydrogels in the future development of soft materials, and what potential impact might this have on various industries?

  8. Is there a specific focus or theme in the review regarding the applications and properties of tough conductive hydrogels, and how is this theme presented throughout the review?

  9. Are there any specific examples or case studies mentioned in the abstract that illustrate the practical implications and benefits of using tough conductive hydrogels in real-world applications?

  10. How does the abstract engage the reader by highlighting the significance of the research findings and their potential contributions to the broader fields of materials science and engineering?

This manuscript reports the Advances in the Preparation of Tough Conductive Hydrogel for Flexible Sensors. In my opinion, the manuscript is suitable for publication in the polymers, after the authors have addressed the following major comments.

  1. How has the rapid development of tough conductive hydrogels impacted the fields of tissue engineering, soft robots, and flexible electronics, and what specific progress has been achieved in each of these domains?

  2. Could you explain the role of the "toughening strategy" within gel networks in regulating mechanical parameters like breaking stress, elongation, modulus, and recoverability? Why are these parameters important for specific applications?

  3. What recent advancements in the preparation of tough conductive hydrogels for flexible sensors are highlighted in the review, and how do these advancements contribute to the field of soft materials?

  4. How does the review emphasize the "toughening strategy" and "conductive method" in the context of hydrogel development, and what makes these aspects significant?

  5. Can you provide examples of applications where these tough conductive hydrogels have been used as flexible sensors with resistive and capacitive modes for detecting additional signals? What advantages do these modes offer in signal detection?

  6. What are some of the current challenges associated with using tough hydrogels as flexible sensors, and how might overcoming these challenges lead to advancements in the development of soft materials for future devices?

  7. Could you elaborate on the future perspectives mentioned in the abstract? How do the authors envision the role of tough hydrogels in the future development of soft materials, and what potential impact might this have on various industries?

  8. Is there a specific focus or theme in the review regarding the applications and properties of tough conductive hydrogels, and how is this theme presented throughout the review?

  9. Are there any specific examples or case studies mentioned in the abstract that illustrate the practical implications and benefits of using tough conductive hydrogels in real-world applications?

  10. How does the abstract engage the reader by highlighting the significance of the research findings and their potential contributions to the broader fields of materials science and engineering?

Reviewer 2 Report

Review of manuscript entitled "Advances in the Preparation of Tough Conductive Hydrogel for flexible Sensors" by H. ding et al.

The manuscript is a review of conductive hydrogels.

Manuscript presents the following sections:

Introduction, Requirements for fabricating tough hydrogels, The design elements for conductive hydrogels, the application of tough conductive hydrogels for flexible sensors and Conclusions.

Introduction shows a view of past work with references, section 3 describes well the electronic and conductive hydrogels, Section 4 describes some applications of the hydrogels, Fig. 5 and Fig. 6 give good examples. In the conclusions section are mentioned the mechanical and electrical characteristics. Long term stability is also treated.

Overall the manuscript has good explanations. It will be good that Fgi. 5 and 6 have a bigger size because in the printed version it is difficult to see the details.

Reviewer 3 Report

This review paper is well-written and provides interesting information on tough conductive hydrogel for flexible sensors. It includes tremendous examples related to the conductive hydrogels and corresponding summary which is well visualized in the table. This paper will be useful guideline for the researchers who are planning to design advanced flexible materials.

Round 2

Reviewer 1 Report

accepted